



# Organizing an Earthquake Learning Exhibition for transferring geoscience knowledge to the public: the example from Nepal

Shiba Subedi[1], Nadja Valenzuela[2], Priyanka Dhami[1], Maren Böse[2], György Hetényi[3], Lauriane Chardot[4], Lok Bijaya Adhikari[5], Mukunda Bhattarai[5], Rabindra Prasad Dhakal[1], Sarah Houghton[6], and Bishal Nath Upreti[1]

[1]Nepal Academy of Science and Technology, Lalitpur, Nepal

[2]Swiss Seismological Service (SED) at ETH Zurich, Zurich, Switzerland

[3]Institute of Earth Sciences, University of Lausanne, Switzerland

[4]Earth Observatory of Singapore, Nanyang Technological University, Singapore

[5]National Earthquake Monitoring and Research Centre, Department of Mines and Geology, Kathmandu, Nepal

[6]St Michael Steiner School, London, United Kingdom

*Correspondence to*: S. Subedi (shibashibani@gmail.com)

**Abstract.** Nepal is located in one of the most seismically active regions on the globe, where a major earthquake is long overdue, yet much of the existing building stock remains highly vulnerable to collapse during intense ground shaking. Public engagement in earthquake preparedness is a vital aspect of reducing casualties and limiting structural damage, with education playing a significant role in shaping both individual and collective protective behaviours. In honour of the 10th anniversary of the devastating, magnitude 7.9 2015 Gorkha earthquake, an Earthquake Learning Exhibition was organized in Pokhara, Nepal, to improve students' knowledge of earthquakes, risk perception, and preparedness. The event showcased fourteen interactive modules that explored earthquake science, causes, and safety measures, engaging nearly 2,000 participants at age 11 to 17. Pre and post-event surveys indicated notable advancements in scientific understanding, with 93% of students identifying plate tectonics as the primary cause of earthquakes, and 95% recognizing their vulnerability to events exceeding a magnitude of 8. Students exhibited increased awareness of structural vulnerabilities, local seismic risks, and the likelihood of experiencing a major earthquake in their lifetime. A significant 85% of those surveyed rated the exhibition positively, with 98% reporting enhanced preparedness, and many indicated plans to share knowledge within their communities, suggesting a ripple effect in disaster preparedness. The exhibition has proven to be an effective and replicable model for integrating interactive learning with community-based preparedness. Recommendations include long-term follow-up and the expansion of teacher training to ensure the sustainability and amplification of its impact.



## 1 Introduction

In regions that are vulnerable to significant seismic threats, public awareness regarding earthquake risks is a fundamental aspect of disaster risk reduction (DRR), vital for fostering community awareness, preparedness, and resilience (UNDRR, 2022). The economic worldwide impact linked to natural disasters, particularly earthquakes, has escalated over the past few decades, with direct losses increasing from an annual average of USD 70–80 billion during the period from 1970 to 2000 to USD 180–200 billion from 2001 to 2020. When cascading effects, including secondary hazards (such as landslides and tsunamis) and disruptions to ecosystem services, are taken into account, the total cost of disasters exceeds USD 2.3 trillion each year (UNDRR, 2025). Earthquakes, characterized by their sudden onset and potential for widespread devastation, significantly contribute to these rising losses, particularly in regions with vulnerable infrastructure and limited emergency response capabilities. These trends emphasize the critical need for proactive DRR strategies that incorporate earthquake-focused public education to interrupt the persistent cycle of seismic disasters, economic losses, and humanitarian crises, further stressed. Indeed, climate change-induced natural and social problems have increased over the past decades, and they are paradoxically competing with solutions for earthquake-caused problems.

Public engagement in earthquake preparedness is a vital aspect of reducing casualties and limiting structural damage, with studies showing that education has a significant impact on both individual and collective protective behaviors (e.g., Ao et al., 2021). Scientists and educators play a crucial role as intermediaries between research and communities, translating complex disaster knowledge into easily understandable information and promoting evidence-based policies aimed at risk reduction (Albris et al., 2020). Education acts as a key instrument in disaster risk management by empowering individuals and communities to assess, prepare for, and respond effectively to hazards, thereby minimizing losses (Scolobig & Balsiger, 2024). Although accurate earthquake prediction is still beyond current scientific capabilities, enhancing preparedness and response strategies through targeted educational initiatives has been recognized as a primary approach to mitigate the impacts of earthquakes, as shown by recent studies in Nepal (Subedi et al., 2020; Hetényi & Subedi, 2023; Maharjan et al., 2023).

Nepal is located in one of the most seismically active areas in the world, where a significant earthquake is deemed long overdue (e.g., Bilham, 2019; Dal Zilio et al., 2021). In spite of this heightened risk, the structural integrity of the current building stock is predominantly insufficient to endure a major seismic event. The National Population and Housing Census 2021 indicates that merely 23% of buildings in Nepal are constructed with reinforced cement concrete (RCC) featuring pillars, a technique acknowledged for its superior seismic resilience. The majority of residences, around 36 %, are built from brick or stone masonry held together with mud mortar, while an additional 24% utilize cement mortar without adequate structural reinforcement (Nepal Census Report, 2023). These traditional and non-engineered construction practices render millions of homes susceptible to collapse during intense ground shaking. Simulation studies predict that approximately 14% of buildings would experience severe damage, whereas total collapse would be confined to about 7% in a scenario



replicating the Mw>8 1934 earthquake with maximum intensity of X scale in eastern Nepal (Chaulagain et al., 2015). Furthermore, forecasts suggest that 50% of buildings could sustain damage and roughly 1.3% of the population could lose their lives in the event of a significant earthquake near Kathmandu; in western Nepal, fatalities could soar to 150,000 under

comparable circumstances (Dixit et al., 2000; JICA, 2002). The 2015 magnitude 7.9 Gorkha earthquake further highlighted these vulnerabilities, resulting in widespread devastation and emphasizing the critical need for enhanced construction standards and disaster preparedness (Shrestha & Hough, 2025).

In light of the significant magnitude of this vulnerability, it is essential to implement immediate and prioritized actions to reduce seismic risk. However, the process of rebuilding or retrofitting millions of residences to comply with earthquake-

resistant standards within a brief timeframe is both economically and logistically impractical. As a result, effective risk reduction strategies must incorporate targeted retrofitting of vital infrastructure, comprehensive community education and awareness programs, along with the advocacy of cost-efficient, resilient construction practices to alleviate potential human and economic losses during future seismic events.

The Seismology at School in Nepal (SASIN) program has been successfully running to enhance earthquake awareness and

for better preparedness since 2017 (Subedi et. al., 2020a). School-based earthquake education programs have demonstrated significant potential to enhance awareness and preparedness in Nepal. For instance, Subedi et al. (2020b) showed that a comprehensive seismology education initiative, integrating classroom instruction, teacher training, and installation of low-cost seismometers, effectively increased earthquake awareness and fostered broader community preparedness. However, altering risk perception remains a complex challenge (Subedi et al., 2020b).

Drawing from these insights, we organized the Earthquake Learning Exhibition 2025 in Nepal to involve students as proactive participants in earthquake knowledge sharing and preparedness. By providing students with scientific knowledge and practical safety techniques, the exhibition seeks to nurture resilient communities that are better equipped to confront future earthquakes. This article outlines the exhibition's design, execution, and results, emphasizing the crucial importance of student-centered education in improving earthquake readiness and promoting community resilience.

The one-day Earthquake Learning Exhibition was held on 25[th] April 2025 in Pokhara, Nepal, to commemorate the 10th anniversary of the 2015 Gorkha earthquake. The event hosted nearly 2,000 students from the surrounding region, primarily from Pokhara Metropolitan City. It featured fourteen interactive modules covering earthquake physics, causes, preparedness, and safety, along with engaging activities such as an artwork competition, a quiz competition, an earthquake awareness song, and evacuation drills. The exhibition was presented by a team of experts, including two international specialists, four

national experts, and school teachers regularly trained by our SASIN program, supported by several volunteer students. To evaluate its impact, surveys were conducted with 500 students before and 309 students after the exhibition, while feedback from students and experts comments were also collected. Through this comprehensive approach, the exhibition aimed to



enhance earthquake awareness and preparedness among youth in an earthquake-prone region, and allowed us to assess the impact of such a short training for a larger group of students.

**2 Exhibition and modules**

The exhibition was held in Pokhara because most earthquake awareness activities in Nepal have historically been concentrated in the capital city region, leaving much of the population outside the capital with limited understanding and preparedness for earthquakes. Importantly, Pokhara and the surrounding western region lie within a recognized seismic gap, where scientists believe a major earthquake is overdue (e.g., Bilham, 2019).


**Module index:**

M1: Plate Tectonics
M2: Magnitude and Intensity
M3: Why we can not predict earthquake
M4: What to do before, during, after earthquake
M 5: BOSS model
M6: Earthquake Artwork Competition
M7: Liquefaction
M8: Seismic Waves
M9: Earthquake Game
M10: Earthquake Emergency Bag (GO Bag)
M11: Seismometer and Jumping Test
M12: Earthquake Quiz Contest
M13: Earthquake Awareness Song
M14: Earthquake Evacuation Drills
R1: Prior-Survey Room
R2: First Aid Room
R3: M14, Post-Survey Room

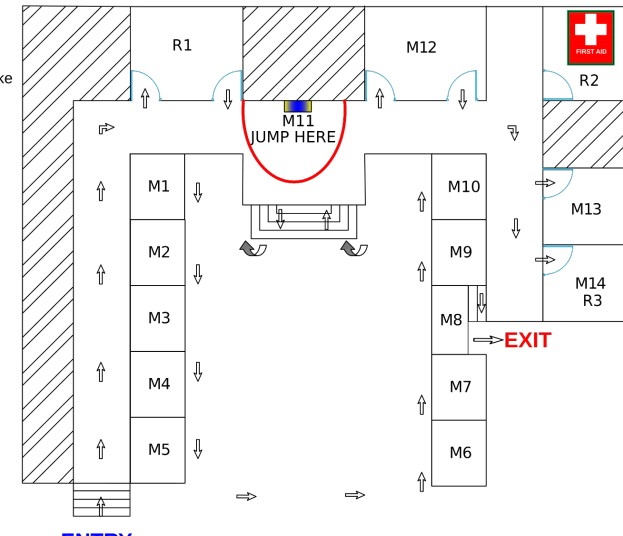

**Figure 1: Exhibition Road Map plot.**



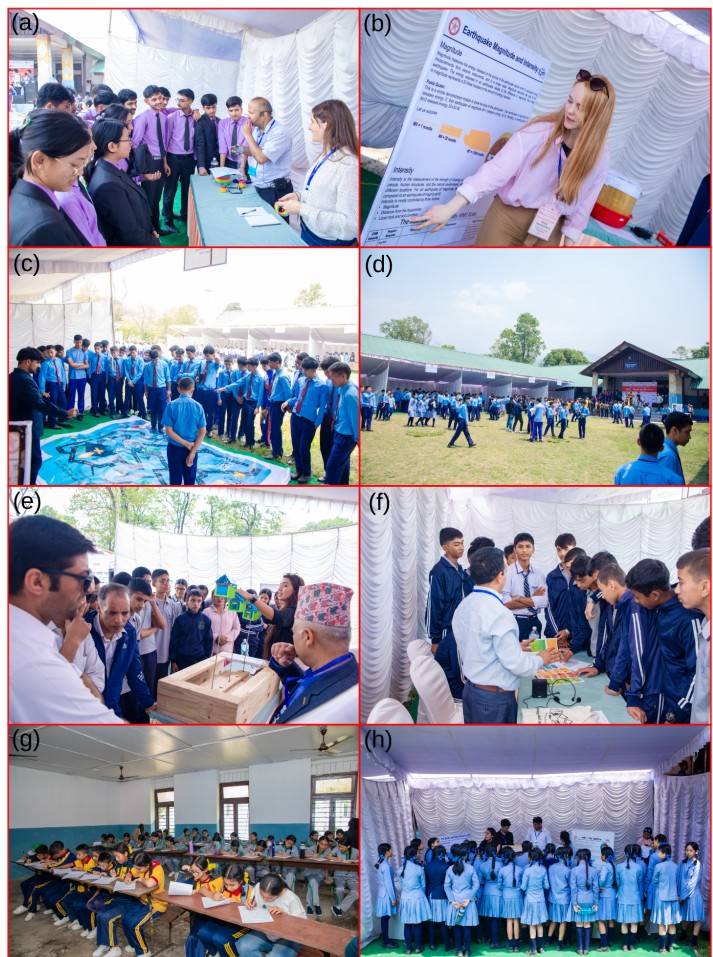

**Figure 2. Example activities during the Exhibition: (a) demonstration of Module 8, seismic waves using slinkies; (b) explanation of Module 2, earthquake magnitude and intensity; (c) students participating in Module 9, earthquake game; (d) students engaging in various modules; (e) students visiting Module 5, BOSS model; (f) explanation of Module 4, preparedness before, during, and after an earthquake; (g) students completing a survey prior to the exhibition visit; (h) students learning Module 1, plate tectonics.**




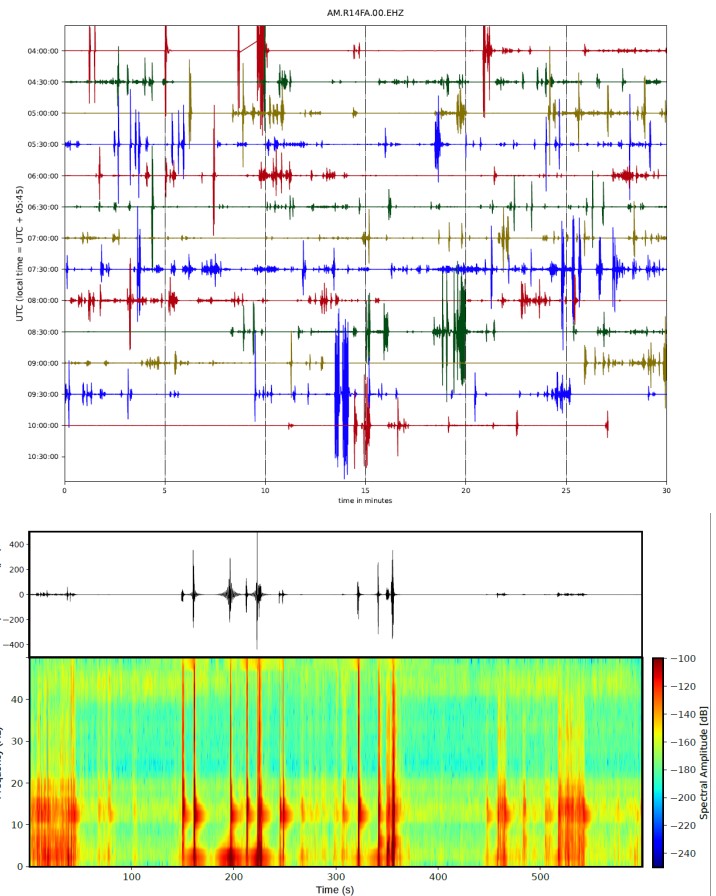

**Figure 3: Seismometer recordings during exhibition activities (top) and jumping tests (bottom).**

The Earthquake Learning Exhibition featured 14 interactive modules designed to educate students on various aspects of earthquake science, preparedness, and response (Figure 1, 2). These modules combined theoretical knowledge with practical demonstrations and engaging activities, ensuring that participants—particularly secondary school students aged 11 to 17 — could gain a well-rounded understanding of earthquake-related topics. Posters prepared and displayed at the exhibition for each module are available in the Supplementary file.



In **Module 1**, which explained the *causes of earthquakes and plate tectonics*, a specially designed mechanical system simulated the continuous subduction of the Indian Plate beneath the Eurasian (Tibetan) Plate. Powered by a motor, this setup produced multiple small slips over time, eventually culminating in a larger slip, which represented a great earthquake. Importantly, the system was connected to a model building on the surface, which visibly shook during these slips, effectively demonstrating how tectonic movements translate into ground shaking and potential destruction. This hands-on visualization

helped students grasp the connection between underground plate movements and surface-level damage during earthquakes.

   In **Module 2**, the concepts of earthquake *magnitude and intensity* were vividly illustrated through innovative yet simple demonstrations. Pasta was used to represent the exponential increase in energy released as earthquake magnitude rises—a convenient and easily understandable method commonly used in Western countries. The demonstration clearly showed that an earthquake just one magnitude unit larger releases approximately 32 times more energy than the previous level, making

this important scientific fact tangible for students. Additionally, a torchlight was used to simulate earthquake intensity, which refers to the shaking level at a given location. The focused beam represented the epicentre, the area experiencing the strongest shaking, while the gradually fading light around it illustrated neighbouring regions that endured less severe shaking and damage. This visual and tactile demonstration clarified the distinction between magnitude (energy released) and intensity (shaking level in a specific area).

**Module 3** focused on the question, *"Why can't we predict earthquakes?"* To demonstrate this, a simple mechanical setup was used, involving elastic rubber, sandpaper, a wooden block, a scale, a clock, and a motor. When the system was activated, the motor pulled the wooden block, storing energy in the stretched rubber. Once the force exceeded the friction between the block and the sandpaper, the block suddenly slipped, mimicking the sudden release of energy during an earthquake. This slip represented a seismic event. The demonstration highlighted that if slips of consistent size occurred at regular time intervals,

earthquake prediction would, in principle, be possible – for example, predicting a magnitude 6 earthquake every 10 years. However, in reality, the size and timing of these slips vary unpredictably. The module thus illustrated the unpredictable nature of seismic activity and why producing a constant, predictable slip pattern is impossible. This hands-on experiment helped participants understand the scientific challenges and current technological limitations that make precise earthquake prediction unfeasible.

**Module 4** focused on *"What to Do Before, During, and After an Earthquake."* This module provided practical, location-based safety guidance reflecting the realities of Nepal's infrastructural environment, where only 23% and 40% buildings are RCC with pillars across the country, and in the given district, respectively (Census, 2021). Participants learned that the commonly promoted "Drop, Cover, and Hold On" technique may not always be effective, especially in non-earthquake-proof structures. Instead, the advice was carefully tailored: if you are outdoors during an earthquake, stay outside and move

to a safe open area away from buildings or other hazards. If you are inside a non-earthquake-proof building on the ground floor, the safest option is to evacuate the building immediately and move outside. For those inside multi-storey buildings,





where exiting quickly is not feasible, the "Drop, Cover, and Hold On" method remains the best way to protect oneself. The module also covered preparation steps such as securing furniture, preparing emergency kits, and pre-planning family communication. This realistic, context-sensitive approach helped students understand how to respond safely depending on
their location and building type during an earthquake.

**Module 5** featured the *Building Oscillation Seismic Simulation* (BOSS) Model, a physical demonstration designed to show how buildings respond to seismic waves during an earthquake. In Nepal, there is a common misconception that taller buildings are more likely to collapse first, while single-story buildings are assumed to be safe. This belief can lead people to construct small, low-rise houses without adhering to building codes, which poses serious risks during strong earthquakes.
For this module, LEGO-shaped blocks were put together without a strong connection of the same design, but different heights were used to represent various building types. Through the BOSS model, participants were introduced to the concept of building response frequency – the natural oscillation rate of a structure – and how it affects a building's response to seismic waves regardless of its height. The demonstration showed that any building, whether short or tall, can sustain significant damage depending on the earthquake's characteristics and the building's design, both functions of frequency. By
visualizing how these structures oscillated during simulated shaking, the module emphasized the critical importance of following earthquake-resistant construction standards to ensure safety across all building types.

**Module 6:** The *Earthquake Artwork Competition* invited students to create artworks on the theme of earthquake awareness and preparedness, using materials such as paint, pencil, and watercolour on A4-sized paper. The artworks were collected prior to the exhibition, and our national and international experts selected the best 15 artworks out of 55, which were later
exhibited at the Earthquake Learning Exhibition. Selected artworks were displayed anonymously at a dedicated booth to avoid bias and ensure fair judgment. A total of 300 students voted for their favourite pieces, and the top three were chosen based on a combined evaluation from both visitors and experts. The winning artworks received certificates, prizes, and cash awards. This process encouraged creativity and actively engaged the community in earthquake preparedness.

**Module 7** focused on *Liquefaction and Earthquakes,* addressing a critical yet often misunderstood phenomenon in Nepal.
Both Kathmandu and the exhibition venue, Pokhara, are built on thick sediment layers, making them potentially vulnerable to liquefaction during earthquakes. While many people may be aware of this risk, the concept is not always fully understood or accepted. This module aimed to clearly demonstrate how liquefaction occurs and why it can cause severe additional damage during earthquakes. For the demonstration, a setup was created using a motor to produce shaking, a wooden base, a transparent plastic box filled with sand and water, and small cars, buildings, and pipes. When the motor generated vibrations,
the saturated sand behaved like a liquid, causing the structures to sink or tilt, effectively illustrating how liquefaction weakens the ground and compromises the stability of buildings and infrastructure. This hands-on visualization helped participants grasp the importance of soil conditions and the need for careful engineering in earthquake-prone sedimentary areas.



**Module 8** explained the nature and properties of *seismic waves*, with a focus on primary (P) and secondary (S) waves and
their behaviour as they travel through the Earth. Participants learned that P-waves are compressional waves that travel the
fastest and arrive first during an earthquake, propagating through solids, liquids, and gases. In contrast, S-waves are shear
waves that travel more slowly, arriving after the P-waves, and can only propagate through solids. To visually demonstrate
these differences, a large slinky was used: compressing and releasing the slinky along its length illustrated the motion of P-
waves, while moving it side-to-side showed the transverse motion of S-waves.

**Module 9** featured an *Earthquake Game* – an interactive, scenario-based activity designed to reinforce earthquake safety
concepts in a fun and immersive way. For this module, a large 12×12 feet colour-printed floor poster of a community map
was used as a game board. The map depicted a fictional neighbourhood that had just experienced an earthquake, showing
damaged buildings, cracked roads, fires, and other hazards. Two participants were placed at different starting points on the
map and were tasked with navigating to a designated safe place or assembly point, choosing the safest available routes while
avoiding blocked paths and danger zones. The activity challenged students to apply their knowledge of earthquake response,
situational awareness, and safe evacuation strategies. It encouraged critical thinking and teamwork under simulated
emergency conditions. The concept was very well received by the participants, who showed enthusiasm and focus in
completing the task. Many successfully followed the correct procedures and made it safely to the assembly point,
demonstrating a strong understanding of what to do after an earthquake. This module not only reinforced key safety
messages but also made learning active and memorable.

**Module 10** focused on the *Earthquake Emergency Bag* (Go-Bag), emphasizing the importance of having a well-prepared,
portable kit for use during and after an earthquake. At the exhibition, a complete Go-Bag was displayed alongside all
individual items, each of which was labelled with its name and purpose to help participants understand their importance. The
bag included water purification items to ensure safe drinking water, drinking water itself to quench thirst, and fast, high-
energy food to provide emergency nutrition. A first aid kit was included to provide immediate life-saving treatment and to
stabilize injuries, while dust masks protected against polluted air. The kit also included essential tools such as a headlamp for
visibility in the dark, a compass for navigation, a whistle for communication and signalling, and a lighter for starting a fire.
Participants were shown notebooks and pens for noting important information, emergency cash for buying supplies when
electronic systems fail, and multi-tools for self-rescue and repairs. Copies of essential documents helped verify identity and
access aid, and a radio kept users updated on ongoing situations. Practical clothing protected against extreme weather and
allergies, complemented by plastic sheets for shelter and rain hats for personal protection. Personal hygiene items helped
prevent infection and disease, and rope was included for rescue and construction purposes. An evacuation map assisted in
navigating risky areas, and family photos aided in reunification and reporting missing persons. Participants were also advised
to regularly check and replace items with expiration dates to ensure readiness. This detailed display provided a practical
example of earthquake preparedness, encouraging families to assemble their own Go-Bags.





**Module 11** focused on *Seismometers and Earthquake Measurement*, offering participants a clear understanding of how earthquakes are detected, measured, and located by scientists. A key component of this module was a detailed explanation of the triangulation method, which uses the arrival times of P-waves and S-waves at different seismic stations to determine the location of an earthquake. To make the learning experience more interactive and relatable, a Raspberry Shake seismometer

was installed at the exhibition venue. Students participated in a fun and educational jumping test, where they jumped near the sensor and observed how their movements were recorded in real time (Figure 3). This activity helped students visualize how energy – similar to that released during an earthquake – is captured as waveforms by seismometers. The hands-on demonstration, combined with real seismic data, made the abstract concept of earthquake measurement more tangible and exciting, and the students thoroughly enjoyed seeing their own activity displayed as seismic signals. For those who are

motivated to locate earthquakes, we suggest following the tutorials explained in Subedi et al. 2021.

In **Module 12** the *Earthquake Quiz Competition* was introduced as a way to indirectly assess how well students had absorbed earthquake knowledge throughout the exhibition. The visit to the different modules was conducted in sequence, from Module 1 to Module 14 (Figure 1). The quiz was scheduled after most of the educational modules had been visited, ensuring participants had ample exposure to the material. Using the Kahoot app on laptops and smartphones, students could

answer questions and see their scores in real-time, adding an interactive element to the experience. To ensure broad participation, selected students from each visiting school took part in the quiz. The questions focused on Earth's interior, tectonic plates, earthquake causes, magnitude, intensity, statistics of earthquakes in Nepal, seismic waves and their velocity, and the formation of the Himalayas. To keep motivation high, the top three scorers were awarded honour prizes, encouraging enthusiasm and reinforcing learning.

In **Model 13**, we used the *Earthquake Awareness Song* (https://youtu.be/ymE-lrAK0TI) to combine entertainment with important safety messages through audio-visual media. The song's video was played continuously while students visited the booth, providing a refreshing and engaging break from the more technical exhibits. Since visitors had been exposed to scientific information for extended periods, this musical module helped maintain their interest and reinforced earthquake preparedness concepts enjoyable and memorable way.

In **Module 14**, we included a practical module on earthquake safety—*Evacuation Drills*—at the exhibition, recognizing that such drills are uncommon in Nepal despite their crucial role in reducing seismic risk in this earthquake-prone country. The module was set up in a room where we first explained the importance of evacuation drills, what participants could expect during the drill, and the correct and incorrect behaviours to follow. Once everyone understood the procedure, we conducted the evacuation from the room to an open space. To simulate a typical Nepali classroom environment, we limited each session

to a maximum of 40 students.



## 3 Survey description

### 3.1 Participants

The participants of the exhibition were primarily secondary school students from public and private schools in the region, mostly from grades 6 to 11, i.e., between 11 and 17 years old (Fig. S10, S11). We visited over 50 secondary schools two

weeks prior and invited them to the event, and in most cases, school principals and teachers welcomed the invitation to participate in this important educational initiative. Additionally, we sent follow-up emails a week before the event with detailed instructions to ensure smooth coordination and preparation for the exhibition. About 88% of the students who attended had experienced either the 2015 Mw 7.9 Gorkha earthquake in central Nepal, or the 2023 earthquakes $M_L$ 6.4 and $M_L$ 6.3 in western Nepal, or both (Fig. S2). Besides students, some teachers and local community members also attended the

exhibition.

### 3.2 Questionnaires

Data for this study were collected using two paper-based questionnaires on the day of the Earthquake Learning Exhibition,

on 25 April 2025. Participants completed the first questionnaire upon arrival at the exhibition and the second one after their visit. Both surveys combined single-answer and multiple-choice questions, with response formats ranging from a single required answer to multiple possible answers (see survey forms in Supplementary File). The questionnaires were available in both Nepali and English.

Both surveys were conducted in a designated room under the supervision of experts and volunteers. Students were allowed

to visit the exhibition only after completing the pre-exhibition survey. Similarly, once they had explored all the exhibition modules, they were asked to complete the post-exhibition survey, which included additional questions to assess the impact of the exhibition. The first survey contained 15 questions, and the second 31 (see supplementary file). Some questions in both surveys were taken from the 2018 and 2020 surveys (Subedi et al. 2020b), to allow longer-term comparisons. The estimated time required to complete each survey was approximately 10 and 15 minutes, respectively. Students who completed the post-

exhibition survey were given a printed educational flyer to take home. In addition, flyers were distributed to representative teachers from each school, with the suggestion to display them on their school's notice boards.

Based on these questionnaires, this study examines the impact of the Earthquake Learning Exhibition on students' knowledge, awareness, and behavioural changes concerning earthquake preparedness. Due to time limitations, a random sample of students was chosen from the approximately 2,000 attendees of the exhibition. A total of 495 students responded

to the pre-exhibition survey, while 309 responded to the post-exhibition survey. Although students were motivated to engage in both surveys, challenges with crowd management resulted in approximately 90% of the same students completing both surveys. Consequently, the responses from the pre- and post-surveys were not matched, as the same individuals did not



necessarily participate in both. The reduced number of responses to the post-exhibition survey was mainly due to participant fatigue, diminishing daylight in the late afternoon, and the advice for school groups to return to their respective institutions

punctually.

For questions that were included in both the pre- and post-exhibition surveys, changes in student responses were evaluated using the chi-square ($\chi^2$) test analysis. The null hypothesis ($H_0$) stated that the exhibition program had no effect on student responses. Rejection of the null hypothesis—indicated by a $\chi^2$ value exceeding the critical threshold for the given degrees of freedom and a corresponding p-value below 0.005—was interpreted as evidence that the program had a statistically

significant impact on students' knowledge, awareness, or behaviour (Table S1). The complete set of survey questionnaires is provided in the Supplement.

### 4 Results

The first question of the survey asked students to describe their initial thoughts upon hearing the word "earthquake". Most responses, about 28%, focused on the physical sensation of shaking, with terms like "earth shaking" and "ground shaking"

being commonly mentioned (Figure 4). Many responses also showed basic understanding of geology, referring to tectonic plates and ground movement. Destruction was a major theme, including mentions of collapsed buildings, damaged homes, and loss of property. Emotional reactions such as fear and worry were frequently noted, alongside references to death and loss. Encouragingly, many students mentioned preparedness actions such as seeking open spaces and using emergency kits. Several responses also referenced the 2015 earthquake, indicating the influence of past experiences. Overall, the answers

reflect a mix of scientific awareness, emotional response, and safety concerns.

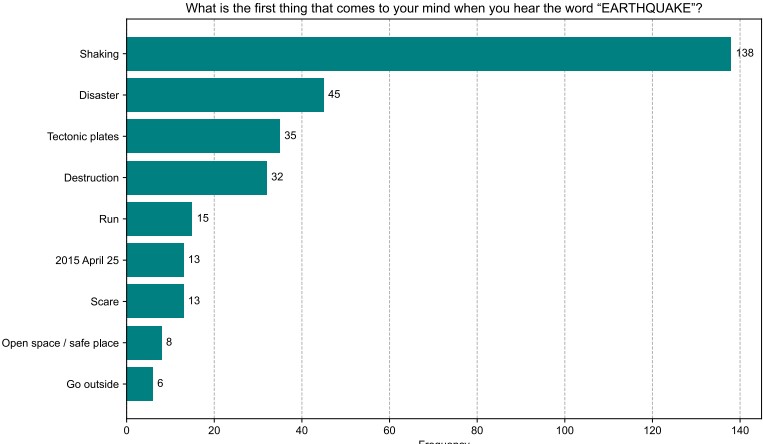

**Figure 4: Student answers to what is the first thing that comes to their mind when they hear the word 'EARTHQUAKE' (Q1).**



The second question of the survey focused on whether participants had experienced earthquakes in recent years, such as the
2015 Gorkha earthquake, the 2023 Western Nepal earthquakes, or any earthquakes at the local level. About 88% of the
respondents indicated that they had felt earthquakes.

### 4.1 Change in Understanding of Earthquake Causes

After participating in the exhibition, students showed a notable improvement in their understanding of the causes of
earthquakes (Q3). More than 93% of respondents identified plate tectonics and scientific factors as the primary causes of
earthquakes, while only 7% attributed them to traditional beliefs and other non-scientific explanations. This positive
development underscores the exhibition's effectiveness, where we specifically included a module that detailed plate tectonics
and the scientific causes of earthquakes. Students not only learned about the tectonic setting of the Himalayas, but they also
explored the comprehensive history of plate tectonics and the movement of the Indian plate with Eurasian over time. In
quantitative terms, there is a ~8% relative increase in the number of respondents who identified plate tectonics as the primary
cause of earthquakes after their visit to the exhibition (Figure 5). Compared to data from an earlier Earthquake Learning
Exhibition in  2018, approximately 29% more students now acknowledge plate tectonics as the cause of earthquakes.

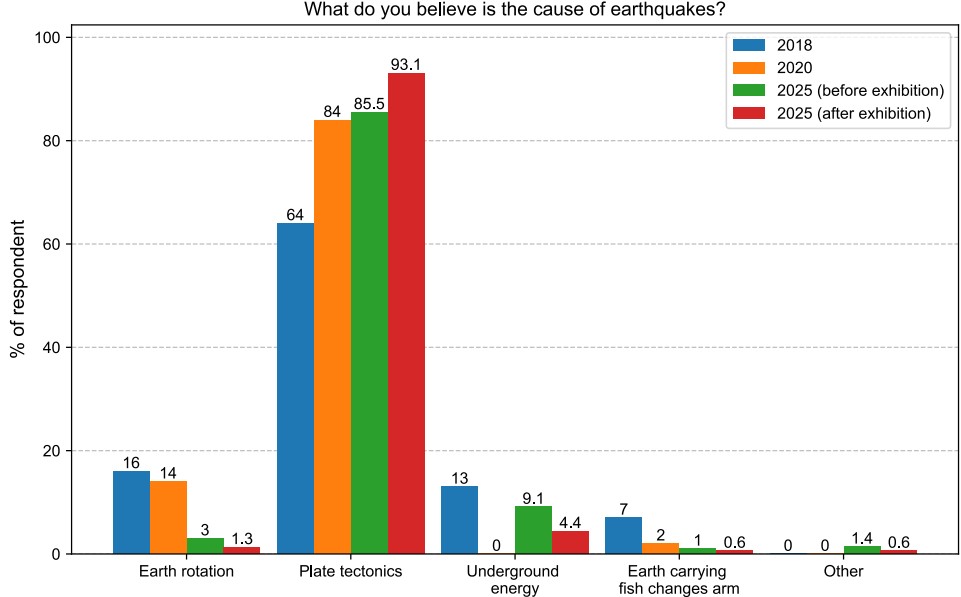

**Figure 5: Students' answers on the causes of earthquakes (Q3), prior and post-exhibition surveys in 2025. ($\chi$2=11.05, p value = 0.026; the change is significant.). The answers from the 2018 and 2020 surveys are also plotted for comparison purposes.**



### 4.2 Change in Earthquake Knowledge and Awareness

Students agree on the fact that a large earthquake will happen in Nepal sooner or later. Approximately 68% of students expect an earthquake exceeding the magnitude of the Gorkha earthquake to occur within their lifetime (Q5). After the exhibition, about 35% of students estimated the likelihood of a major earthquake in Nepal as 70% or more, which is approximately 8% higher than the survey results before the exhibition. More importantly, only less than 2% said it is impossible to have a large earthquake in Nepal in their lifetime (Figure 6). This shows that they have gained knowledge on seismic rates and their records in Nepal throughout history.

More than 90% of students reported having discussed earthquakes in their classrooms (Q11), which aligns with the fact that a similar proportion had personally experienced an earthquake (Q2). This is expected, as felt earthquakes often prompt discussions, especially in educational settings.

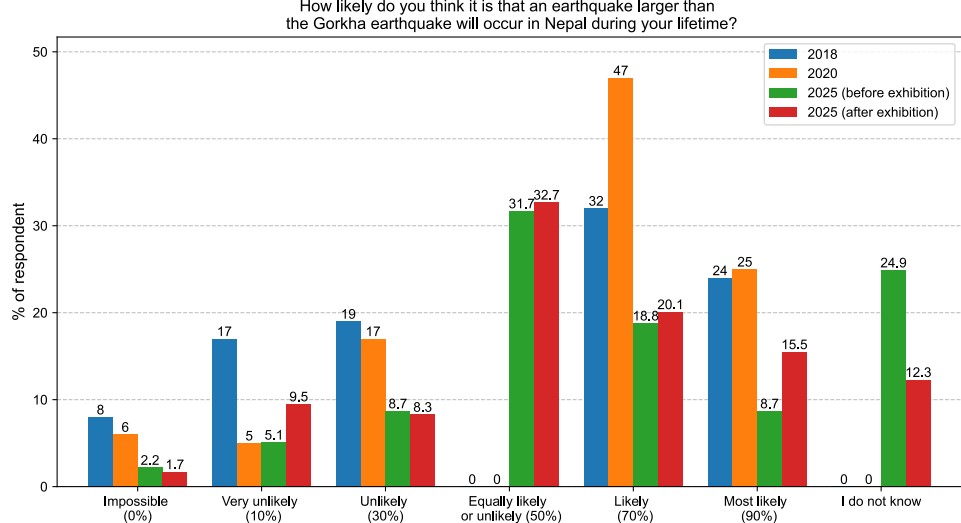

**Figure 6: Responses on the possibility of an earthquake larger than the 2015 Gorkha earthquake in their lifetime (Q5). ($\chi2=30.83$, p value = 0.00002; the change is significant.). The answers from the 2018 and 2020 surveys are also plotted for comparison purposes. Note that an 'equally likely or unlikely' and 'I do not know' option was added in the 2025 surveys.**

Student participation in disaster risk education training and related events has shown a notable increase over the last couple of years, mainly after the Gorkha earthquake. About 13% more participants reported having taken part in disaster risk education programs, likely including such programs by the school, local government, or the exhibition itself, in the 2025 post-exhibition survey compared to the pre-exhibition survey. When compared to data from 2018 (Subedi et al., 2020b), participation had increased by over 200%, reflecting growing awareness and engagement among students (Figure 7).




After the exhibition, respondents showed increased knowledge about earthquake-prone areas and local government contact information. In particular, students' knowledge has significantly improved — in 2025, about 35% more students know the earthquake-prone areas compared to 2018, and about 21% more students are aware of open spaces where to go in case of earthquakes after the exhibition visits (Q14). Additionally, the number of participants who believed that the exact timing of

an earthquake could be predicted decreased by 18% in 2025, indicating improved understanding of earthquake science, which is supported by over 50% students participating in disaster risk-related education training (Q12) (Figure 7).

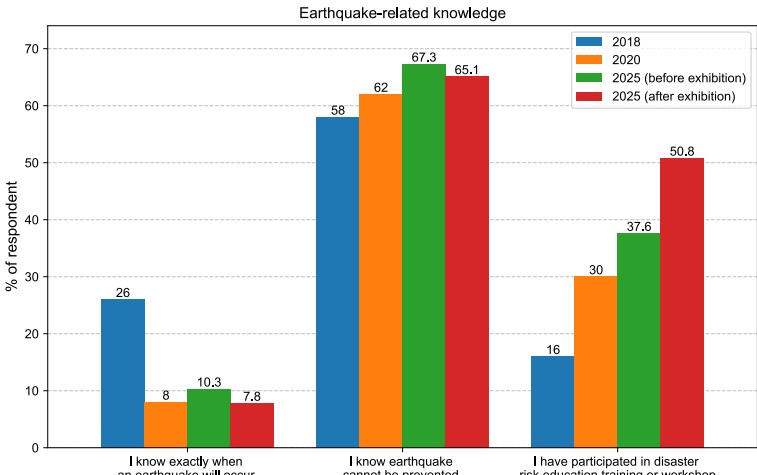

**Figure 7: Students' knowledge about earthquakes before and after the exhibition (Q12). The answers from the 2018 and 2020**
**surveys are also plotted for comparison purposes. (For this question, multiple answers were allowed.)**

Although the exhibition repeatedly emphasized that phone calls should be avoided immediately after an earthquake to preserve lines for emergency use, many students still indicated a preference for contacting family and friends in the event of a major earthquake (Q8). This suggests that they may not fully understand the risk of network overload or outages during

large disasters. The tendency to prioritize communication likely comes from emotional concern, highlighting the need for more education on safe and practical ways to communicate during emergencies.

**4.3 Change in Earthquake Preparation**

There have been positive changes in earthquake preparedness following the exhibition. The number of respondents who believe that survival in an earthquake depends solely on luck has decreased to around 24% from 30% (Figure 8). At the same

time, the perceived importance of sharing earthquake-related knowledge and experiences, as well as discussing hazards





within the community and with family and friends, has been steadily increasing. Notably, the percentage of respondents who recognize the importance of sharing earthquake knowledge and experiences rose consistently from 21% in 2018 to 72% in 2025, and the importance of talking about the hazards within the community rose to 57% in 2025, as it was 31% in 2018 (Q13). However, after the exhibition, students expressed slightly less trust in both government-led reconstruction efforts and the reliability of government facilities following a disaster (Figure 8).

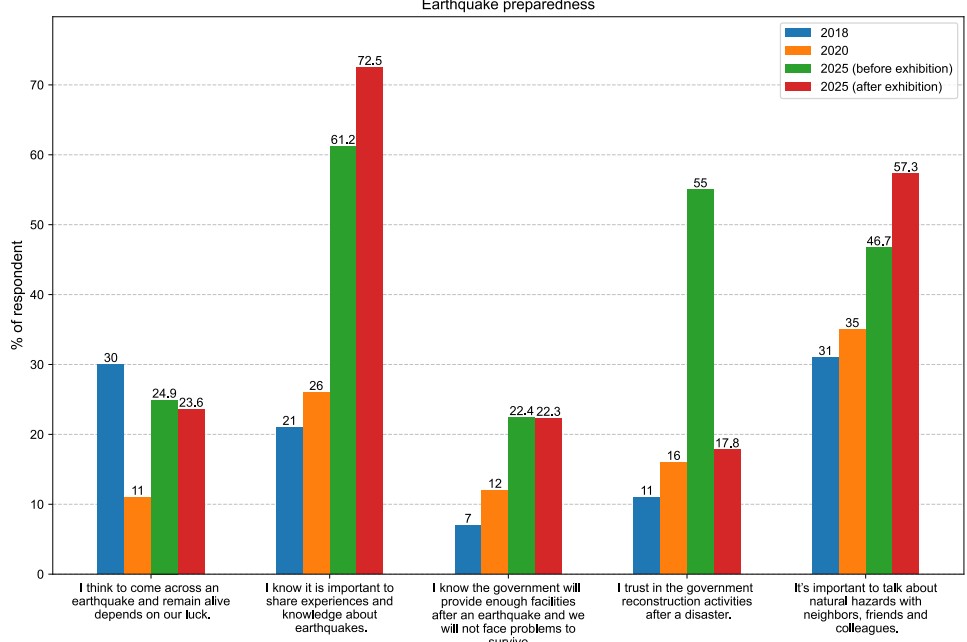

**Figure 8: Students' responses on earthquake preparedness before and after the exhibition (Q13). The answers from the 2018 and 2020 surveys are also plotted for comparison purposes. (For this question, multiple answers were allowed.)**

Over half of the participants reported that they discussed the topic of earthquakes with their families. This number increased by 5% after visiting the exhibition (Q10). Since the exhibition did not cover this topic, the slight increase in responses appears unrelated.

**4.4 Change in Risk Perception**

In our earlier surveys in 2018 and 2020, risk perception was more difficult to appreciate earthquake-related educational activities (Subedi et al., 2020b). Interestingly, for this event, we found that students' risk perception changed significantly after attending the exhibition. Over 95% of students indicated that they would be at risk if an earthquake with a magnitude





greater than 8 occurred in the region (Q4). This heightened sense of risk is further reflected in the increased number of students who responded "my home could collapse" in the post-exhibition survey, demonstrating a clear shift in their perception of personal vulnerability. Additionally, approximately 80% of students anticipate that an earthquake exceeding

the magnitude of the 2015 Gorkha earthquake will occur within their lifetime (Q5).

About 50% more participants shifted their risk perception after visiting the exhibition, indicating that they now believe the risk level in their region is high. However, more than 70% of participants still perceive the risk as medium (Figure 9). This may be attributable to the fact that many respondents reside in reinforced cement concrete (RCC) buildings and/or have never experienced a damaging earthquake, and that the explanations provided regarding the associated risks were insufficient

to fully persuade them (Q9).

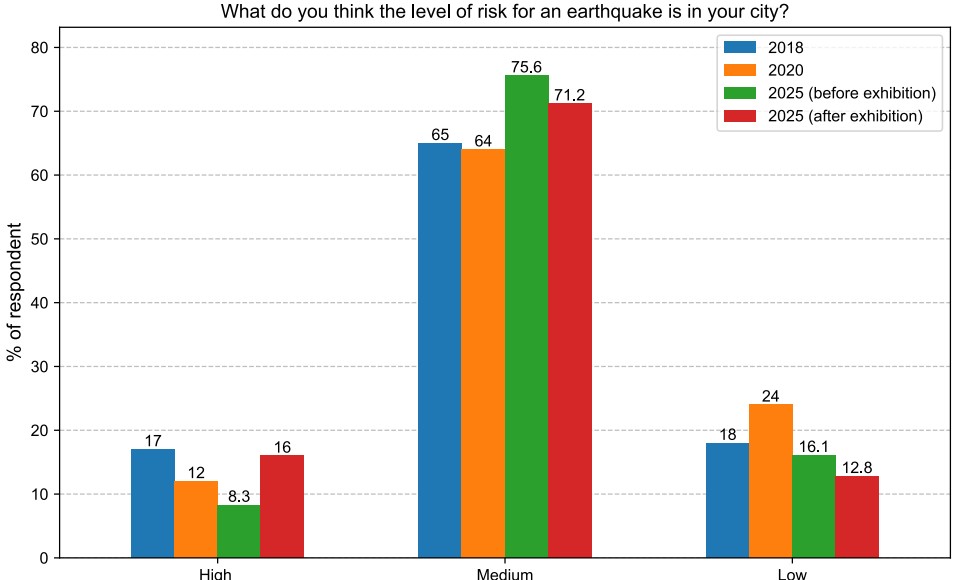

**Figure 9: Students' earthquake risk perception before and after the exhibition (Q9). (χ2=12.88, p value = 0.00156; the change is significant). The answers from the 2018 and 2020 surveys are also plotted for comparison purposes.**

**4.5 Impact of the Exhibition**

We have included several questions in the post-exhibition survey to measure the success of the Earthquake Learning Exhibition. These questions were designed to evaluate both the educational value and the practical impact of the exhibition. The survey explores which aspects participants found most beneficial and how effectively the content enhanced their



knowledge and preparedness for earthquakes. It also investigates whether the exhibition influenced participants' perceptions

and attitudes toward earthquake safety, and whether they now feel more confident in their ability to respond to such emergencies. Additionally, the questions assess the likelihood that attendees will recommend the exhibition to others and share their newly acquired knowledge within their communities. Participants are encouraged to reflect on how they intend to disseminate this information, identify key individuals in their community who could benefit from it, and suggest any additional resources that could support their awareness efforts. Finally, the survey seeks to understand whether attendees are

motivated to take concrete preparedness actions, such as creating emergency plans, securing household items, or promoting earthquake safety, following their experience.

It was noted that nearly all students found the exhibition highly beneficial, particularly for learning about earthquakes. Specifically, 45% of students highlighted the value of understanding general earthquake concepts, 41% appreciated learning the appropriate actions to take before, during, and after an earthquake, 22% found the information on seismic activity in

Nepal to be the most useful, and 15% emphasized the importance of gaining insight into the causes of earthquakes (Q17).

When asked to rate the effectiveness of the exhibition in improving earthquake knowledge and preparedness, students expressed high levels of satisfaction with both the exhibition and its modules. Approximately 85% of participants indicated that the program was effective, while only 4% considered it ineffective (Q18) (Figure 10).

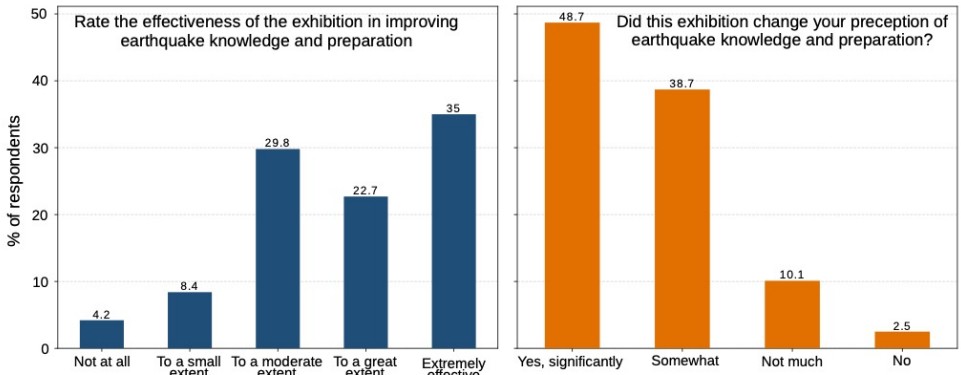

**Figure 10: Impact of the exhibition. (left) Students' satisfaction statistics after the exhibition (Q18). (right) Students' change in perception regarding earthquake knowledge and preparedness (Q19).**

According to the responses, 87% of the students who attended the exhibition reported a change in their knowledge and perception regarding earthquakes (Q19). Notably, nearly 50% of them stated that the exhibition significantly changed their

perception (Figure 10).

In response to the open-ended question "Which new information or skills did you learn from this exhibition?", students most often mentioned practical safety measures like preparing an emergency bag and using the "Drop, Cover, Hold" technique (90




mentions). Many also highlighted learning about what to do before, during, and after an earthquake (70 mentions), tectonic plates and the causes of earthquakes (60 mentions). Other common themes included seismic waves and tools such as

seismometers (50 mentions), general earthquake facts (40 mentions), preparedness strategies (25 mentions), first aid and rescue skills (20 mentions), and the BOSS model (25 mentions), 15 students said they learned nothing new, while a few mentioned engaging tools like an earthquake game (8 mentions) and other topics such as mountain formation (10 mentions) (Figure 11).

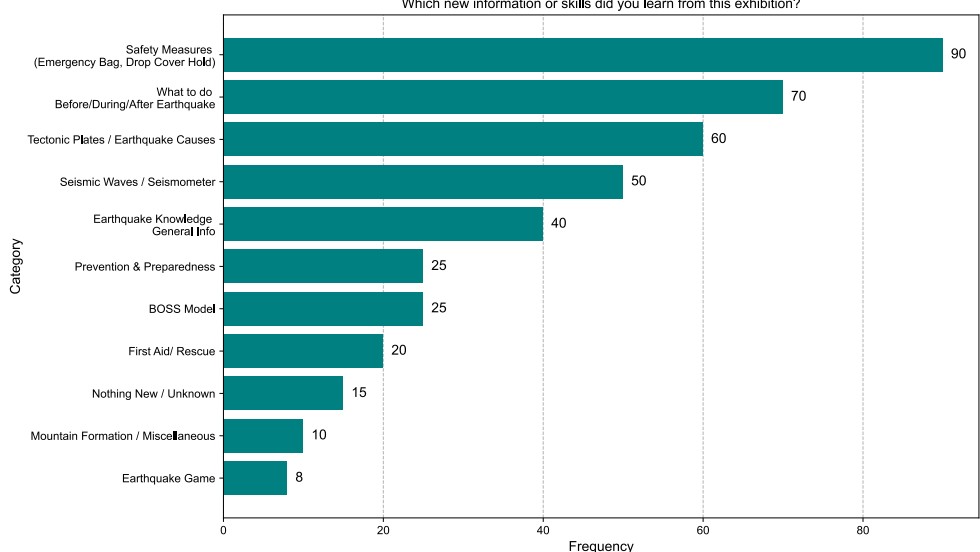

**Figure 11: Students' answers about the new information learned from the exhibition (Q20).**

Students' satisfaction with the exhibition stems not only from the content presented but also from how they felt afterward. When asked about their increased confidence and preparedness to handle an earthquake, 65% of students answered "Yes", 33% responded "Somewhat", and only 1.9% said "No". These results suggest that most participants felt more confident in

their ability to respond to an earthquake after attending the exhibition. As a result of the numerous demonstrations and explanations presented at the exhibition, 98% of students feel better prepared for future earthquakes (Q21).

To ensure the knowledge gained through the exhibition reaches a broader audience, it is important that students share what they've learned with their communities. However, this does not always happen in practice. Overall, 68% of students indicated that they would recommend the exhibition, as they found it effective for learning about and preparing for

earthquakes (Q24).




Students plan to share their exhibition knowledge with community members, underscoring the role of social networks in spreading disaster preparedness. Over 90% of students reported that they intend to share earthquake-related knowledge within their communities. Nearly 60% said they would share with more than 10 people, reflecting both high levels of engagement and the strength of their personal networks. Based on these responses, it is estimated that around 2,400

community members will receive earthquake-related information directly from the students (Q25) (Figure 12). When asked how they would do this, 68% mentioned they would talk directly to family and friends, 30% intended to use social media, 15% planned to teach what they learned in schools or universities, and 24% aimed to organize community discussions (Q26) (Figure 12b).

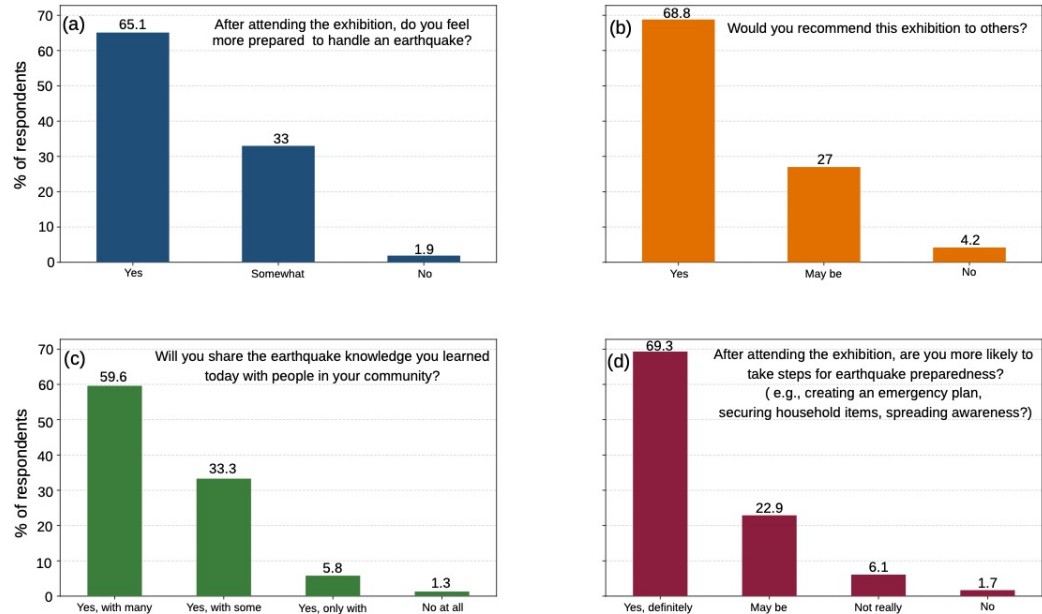

**Figure 12: Impact of the exhibition. (a) Students' answers on their feelings for preparation to handle an earthquake (Q21). (b) Student's recommendation of the exhibition to others (Q24). (c) Student's interest in sharing the knowledge with the community (Q25). (d) Students' answers on whether they would take steps for earthquake preparedness after visiting the exhibition (Q30).**

While students found the information important and helpful for themselves, they also recognized its relevance beyond their
peer group; many identified both school-aged children and senior citizens as key beneficiaries of earthquake preparedness education (Q27). Every student who completed at least one of the surveys received an educational flyer. These materials appeared to be quite beneficial, as half of the students indicated that supplementary educational resources could further



enhance their awareness of earthquakes. Additionally, 60% expressed interest in receiving further support from professionals or the local government through in-person or online events (Q28).

The post-exhibition feedback highlights strong appreciation for the Earthquake Learning Exhibition, with many describing it as "very helpful," "useful", and "inspiring". Numerous respondents praised the educational quality, reflecting the exhibition's success in raising awareness. While some suggestions were made to improve comfort and logistics, such as better amenities and crowd management, these offer valuable opportunities for enhancing future events. Overall, the feedback demonstrates high satisfaction and enthusiasm for the exhibition's impact on earthquake preparedness and

education (Q29).

As a result of the successful exhibition, nearly 70% of students indicated that they are likely to take steps toward earthquake preparedness (Q30), demonstrating the event's strong motivational impact (Figure 12d). The exhibition encouraged students to take initiative, with many planning to apply what they learned in meaningful ways. Among the actions they are most likely to take, educating others stands out as the most important. About 53% of students intend to share earthquake safety

knowledge with their families and friends, while approximately 31% plan to prepare emergency kits. Additionally, 17% aim to learn more about earthquake-resistant buildings, and another 18% are interested in participating in community engagement efforts to promote earthquake safety (Q31).

## 5 Discussion

The Earthquake Learning Exhibition in Pokhara, Nepal, organized to mark the 10th anniversary of the 2015 Gorkha

earthquake, revealed substantial progress in students' knowledge about earthquakes, their risk perception, and preparedness. The exhibition's interactive design, which included 14 modules addressing earthquake physics, causes, and preparedness strategies, facilitated a significant shift in scientific understanding. For instance, the fact that 93% of students identified plate tectonics as the primary cause of earthquakes demonstrates an improved comprehension of key geoscience concepts. This knowledge is vital for nurturing informed attitudes towards seismic hazards and for building safer communities.

The exhibition played a crucial role in shaping students' views on personal and structural risk. After attending the exhibition, 95% of students acknowledged being at risk from earthquakes reaching or exceeding a magnitude of 8, and the marked decline in the belief that their residences could withstand such seismic events points to an increased understanding of building vulnerabilities and structural safety issues. This is in line with research that demonstrates the effectiveness of experiential learning and interactive displays in enhancing risk awareness and promoting protective behaviors.

In addition to personal comprehension, the event promoted awareness regarding risks at the community level. More students recognized their city as significantly vulnerable, and 80% expected to face a major earthquake during their lifetime, highlighting the exhibition's effectiveness in communicating local seismic risk factors.

Feedback from students underscored the exhibition's effectiveness and engagement, with over 85% rated it favourably, and 98% reported feeling better prepared for earthquakes. Importantly, the exhibition also inspired community-driven actions, as



the majority of students showed a strong inclination to share earthquake awareness with their families, friends, and social networks. Notably, about 31% students plan to prepare an earthquake emergency kit, which is crucial for saving lives after an earthquake. This ripple effect illustrates the potential for educational programs to influence not just direct participants but also to cultivate a culture of preparedness in wider communities.

Still, certain operational and logistical hurdles, like crowd management and resource distribution, need to be resolved to
optimize upcoming events. In addition, the sustained retention of knowledge and the implementation of knowledge into ongoing preparedness initiatives remain topics for further research.

## 6 Conclusions

The Pokhara Earthquake Learning Exhibition significantly improved students' comprehension of scientific concepts, their awareness of risks, and their preparedness behaviors concerning earthquakes. Interactive, student-centered educational
activities not only strengthened individual and collective awareness but also encouraged proactive readiness. Although there is a necessity for operational enhancements and an evaluation of long-term effects, the exhibition illustrates a model that can be replicated for merging interactive learning with community preparedness, offering potential advantages that reach far beyond the immediate participants.

## 7 Recommendations

Given its success, the Earthquake Learning Exhibition serves as a model for scalable and community-driven earthquake education initiatives across the country. To build on this momentum, it is recommended to:

- Conduct regular follow-up programs to evaluate long-term impact and reinforce key messages.
- Expand teacher and volunteer training to ensure consistent and effective delivery of content.
- Collaborate with local governments and schools to institutionalize earthquake education programs.
- Develop additional educational materials (physical and digital) for distribution purposes.
- Organize community-inclusive workshops, encouraging student participation as co-educators.
- Address logistical and organizational challenges to ensure a smooth and impactful event experience.
- Refine and implement earthquake education policy, for which a proposal has been put forward (Hetényi and Subedi, 2023).



**Data availability**

The data sets used for this study can be made available from the corresponding author upon request.

**Supplement**

The supplement related to this article is available online at …


**Author contributions**

All authors contributed to the study conception, design, and material preparation. S. Subedi prepared all the figures and the first draft, and all authors contributed to the manuscript. All authors read and approved the final manuscript.

**Competing interests**

The authors has declared that they have no competing interests.

**Ethical statement**

This study was conducted with consent from the head of school, adhering to ethical guidelines. Any personal data collected was anonymized, and participation was voluntary, prioritizing students' educational and safety benefits.

**Acknowledgements**

The authors gratefully acknowledge the Nepal Academy of Science and Technology (NAST) for organizing the exhibition.
We are especially thankful to Seismology at School in Nepal (SASIN) for their technical, volunteer, and content support, as well as for their collaboration throughout the event. We also acknowledge the UNIL-TU NEPALPINE project for supporting the international two-day workshop on educational seismology for teachers, which allowed teachers to train on the modules. We extend our sincere thanks to Kushal Pokhrel, Shahil Sharma, Shreekanta Subedi, Hari Ram Shrestha, Kabita Pandey, and Samundra Kandel for their invaluable assistance in preparing and conducting the exhibition. Our appreciation further goes to
the volunteer teachers trained by SASIN and the student volunteers from Prithvi Narayan Campus for their active contributions. The authors are also grateful to the SED ETH Zurich (Seismology at School in Switzerland) project and EOS Singapore for facilitating the participation of Nadja Valenzuela and Lauriane Chardot, respectively. Finally, we acknowledge the support of Pokhara Metropolitan City and all the participating schools.

**Financial support**
This work has no additional financial support.



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
