# Peer review of "Organizing an Earthquake Learning Exhibition for transferring geoscience knowledge to the public: the example from Nepal"

_EGUsphere, 2025_

## Author Comment (AC1)

**Organizing an Earthquake Learning Exhibition for transferring geoscience knowledge to the public: the example from Nepal**

Shiba Subedi, Nadja Valenzuela, Priyanka Dhami, Maren Böse, György Hetényi, Lauriane Chardot, Lok Bijaya Adhikari, Mukunda Bhattarai, Rabindra Prasad Dhakal, Sarah Houghton, and Bishal Nath Upreti

**Answer to reviewers**

**Reviewer #1:**

This is an excellent manuscript, and I want to begin by congratulating you on the organisation of such an impactful event. The Pokhara Earthquake Learning Exhibition is a remarkable achievement — reaching nearly 2,000 students with 14 carefully designed, interactive modules is no small task. The manuscript communicates both the scientific content and the community value of this initiative very effectively. The scale of engagement, the thoughtful adaptation of global earthquake education methods to the Nepali context, and the integration of science, art, and practice are all commendable. I particularly appreciated the creative modules (e.g., the BOSS model, liquefaction demonstration, earthquake game) and how these were connected to local realities such as non-engineered buildings, sedimentary basins, and context-specific safety advice. This paper highlights a model that has the potential to inspire similar initiatives worldwide.

That said, there are a few areas where the paper could be strengthened. the study already makes a significant contribution, and clarifying some methodological and interpretative aspects will make it even stronger.

We are very grateful for this kind and careful review. We revised the manuscript according to these points, as well as a few other polishes, and provided the respective answers to each question/concern below.

**Key suggestions**

The pre- and post-exhibition surveys were not matched at the individual level, yet the Results are sometimes written in a way that implies causality (e.g., "the exhibition improved awareness" or "students gained knowledge"). In reality, the analysis compares two partially overlapping groups, and differences may partly reflect sampling bias (e.g., more motivated students remaining for the post-survey) rather than direct learning effects. It may be helpful to soften causal language and add a clear note in the Methods and Discussion to acknowledge this limitation.

Thank you for the comment. Although the pre- and post-exhibition surveys were not matched at the individual level, over 90% of respondents who participated in the post survey had already participated in the pre survey. We have added a clear note in the Discussion section about this and the limitation that it is not 100%.

From ~2,000 attendees, 495 pre and 309 post responses were collected. It would strengthen the paper to discuss whether those who completed the post-survey may have been more motivated or engaged, and how this might affect interpretation.

Thank you for the comment. We have added a sentence in the Questionnaire section 3.2. We suspect that if we could collect the same number of responses in the pre- and post-surveys, it would positively change the results, since the post survey may have been more motivated and engaged. We assume there should be a negligible effect on our interpretation.

In Module 3 the statement "predicting a magnitude 6 earthquake every 10 years" could be misread as a realistic claim. Please clarify that this is a hypothetical illustration only.

We have removed the above-mentioned statement.

Line-by-line comments

L15: "on the globe" → "of the globe"

The word is corrected.

L19: "magnitude 7.9 2015 Gorkha earthquake" → "the 2015 Mw 7.9 Gorkha earthquake"

The phrase is corrected.

L27: Please use "pasta sticks" or "uncooked pasta strands" for clarity

We have not seen pasta-related words or sentences in L27.

L41: "further stressed" seems like a leftover phrase and interrupts the flow.

The word further stressed is changed to further highlighted.

L91: The Introduction ends with details on survey numbers (500 pre, 309 post). These figures fit better in Methods. Keep a short mention that impact was assessed with pre/post surveys, but move the numbers.

We have removed the survey numbers from the introduction but kept the survey information.

L239: "preparedness concepts in an enjoyable and memorable way" — add "in an" before "enjoyable."

We have added 'in an' before enjoyable.

L250–253: Clarify how the 50 schools were chosen — was it random, or based on proximity to Pokhara? Were private and public schools proportionally represented?

We have clarified how the 50 schools were selected. We also added additional information about the invited schools.

L267: "The first survey contained 15 questions, and the second 31" → Please explain why the second survey was longer, and whether this influenced completion rates.

We have clarified why there are more questions in post post-exhibition survey, which is because of adding questions for the impact assessment of the exhibition. The information is updated in the manuscript.

Consistently use "post-exhibition survey" rather than "post survey."

We have used post-exhibition survey in the manuscript.

"Model 13" should be corrected to "Module 13."

The word 'Model" is corrected to Module.

---

## Author Comment (AC2)

**Organizing an Earthquake Learning Exhibition for transferring geoscience knowledge to the public: the example from Nepal**

Shiba Subedi, Nadja Valenzuela, Priyanka Dhami, Maren Böse, György Hetényi, Lauriane Chardot, Lok Bijaya Adhikari, Mukunda Bhattarai, Rabindra Prasad Dhakal, Sarah Houghton, and Bishal Nath Upreti

**Answer to reviewers**

**Reviewer #2:**

This manuscript summarizes the outreach activities carried out during an "Earthquake Learning Expo" held in Nepal and analyzes the results of a survey in which participants were asked about various aspects related to seismic activity and risk. The topic is of interest as it could serve as an example for exporting this type of activity to other regions of the planet, especially those prone to large earthquakes.

The manuscript is well written and, in my opinion, complies with the editorial policy of Geoscience Communication. Therefore, I recommend its publication in this journal, although I also recommend modifying some aspects.

Thank you very much for your constructive review. We are very grateful for the careful review. We revised the manuscript according to these points, as well as a few other polishes, and provided the respective answers to each question/concern below.

My main concern is focused on a thorough review of the survey results. Given the small sample size (a few hundred), I do not believe it is relevant to analyze the percentages of responses to each question in detail. Furthermore, Section 4 is closer to a technical report than an article on scientific communication. I recommend selecting only a few questions and figures that illustrate the key points, i.e., the improvement of knowledge about earthquakes and seismic risk. The remaining material could be included as part of the Supplementary Material.

In the context of this Exhibit, we believe that a sample size of a few hundred people is representative and adequate, and we prefer to keep reporting response percentages for each question. We agree with the Reviewer to present a smaller set of key questions in the manuscript and include the remaining ones in the supplementary materials. Accordingly, we decided to remove Figures 1, 4, 9, 10, and 11 from the main text while retaining them in the supplementary section.

Furthermore, I believe the section describing the activities carried out in the different modules of the exhibition could be improved, for example, by including images showing the different devices used, as these could be of interest to those planning this type of event. Furthermore, I would appreciate reference to research resources that the authors have likely referenced, such as educational materials provided by the USGS, Raspberry Shake, or others.

Because all modules were developed in Nepal using locally available materials, the specific components may differ for users in other regions; however, the underlying principles remain the same. Presenting all the items that are used in the module makes the manuscript long, and if we use a photo of each module, it is hard to show all the items in a single picture. We have added the following sentence to the Data Availability Statement: 'Anyone interested in

replicating these modules may contact the authors for detailed information on the materials used in each module.' We have listed all the links that inspired us to make these modules and cited all of them.

Overall, I have the impression that the manuscript gives more emphasis to the survey than to the exhibition activities, while, in my opinion, what is most relevant is precisely the exhibition itself.

We have well described the exhibition, including the modules and activities we have performed in the exhibition. We also presented results from the surveys on the impact of the exhibition. We prefer to keep both aspects in the manuscript.

Below I include some second-level observations and comments: Introduction

The introduction, and especially the second paragraph, includes generic phrases that provide no scientific information and, in my opinion, are unnecessary.

The introduction section aims to inform the readers about the importance of earthquake preparedness in terms of the current risk level. The second paragraph reflects the importance of education in earthquake risk reduction. As the exhibition is spreading education to the public for the same reason, we decided to keep the paragraph in the introduction.

When discussing the audience attending the exhibition, I recommend including the description now in subsection 3.1, as it provides information on the number of schools, the ages of the participants, etc.

The description of the audience attending the exhibition is now mentioned in subsection 3.1. Thank you for noticing this.

**Section 2**

Figure 1, which shows the floor plan of the exhibit, is not really necessary. We have moved Figure 1 from the manuscript to the supplementary material.

Figure 2: It would be easier for the reader if the photos were sorted by module number. We have sorted the photos by module number.

Figure 3: The time interval corresponding to the spectrogram could be indicated in the top panel.

We have accepted the comment. Figure 2 is modified accordingly.

As mentioned above, I think including photos, plans, or sketches of the devices used could be interesting for the reader.

I understand that many of these devices and activities have been inspired, directly or indirectly, by educational materials available online. I think it would be appropriate to explicitly reference them.

We have listed all the links that inspired us to make these modules and cited all of them. The references we have used are mentioned as follows:

Module 2:https://www.iris.edu/hq/inclass/activities/magnitude\_and\_intensity (torch), https://www.iris.edu/hq/inclass/activities/pasta\_quake\_exploring\_earthquake\_magnitude (pasta quake).

Module 5:

https://www.iris.edu/hq/inclass/lesson/demonstrating\_building\_resonance\_using\_the\_simplified\_boss\_model, the original BOSS Model is from Ireton et al., 1995

Module 6: https://www.youtube.com/watch?v=BxtiKodKq\_E

Module 8: https://web.ics.purdue.edu/~braile/edumod/slinky/slinky4.pdf

Module 10: https://www.usgs.gov/fags/what-emergency-supplies-do-i-need-earthquake

Module 11: https://www.usgs.gov/media/images/triangulation-locate-earthquake,

https://www.iris.edu/app/triangulation/

As a matter of style, beginning each paragraph with "Module xx" is more appropriate in a technical report than in an article. However, I understand that it is probably the most efficient way to describe the content of each module.

We believe that the current presentation format is the most effective for describing the content of each module and guiding readers through the material.

**Section 3**

When using "(Q3)" for the first time, explain that it refers to "question 3" and that its exact wording can be found in the supplementary material.

Thank you, we changed Q3 to question 3 in the manuscript. We keep the consistency of this style for each question.

As mentioned above, I recommend drastically reducing the number of figures in this section, which now includes nine figures, all of which show bar charts.

We agreed to move five figures from the manuscript to the supplementary material.

**Conclusions**

I recommend merging the Discussion, Conclusions, and Recommendations sections into a single section, as the current Conclusions section consists of only two sentences and six lines.

Along the same lines as the previous comments, I believe ending the manuscript with a bulleted list of recommendations is more appropriate for a report than for an article in a journal like Geosciences Communication.

We agree with the reviewer on merging the three sections. The 7 bullet points are converted to text, in three parts: what concerns students and participants, what concerns teachers and future educators, and what concerns organization, politics, and policy. For each of these, we still used (i) and (ii)-format indications of items within a sentence. The text is updated accordingly.

---

## Author Comment (AC3)

**Organizing an Earthquake Learning Exhibition for transferring geoscience knowledge to the public: the example from Nepal**

Shiba Subedi, Nadja Valenzuela, Priyanka Dhami, Maren Böse, György Hetényi, Lauriane Chardot, Lok Bijaya Adhikari, Mukunda Bhattarai, Rabindra Prasad Dhakal, Sarah Houghton, and Bishal Nath Upreti

**Answer to reviewers**

**Reviewer #3:**

This is a solid manuscript presenting surveys and stats associated with a fantastic outreach and education event. It is well worthy of publication, has international relevance, but some key things need to be tidied up for stronger presentation and discussion, especially around clarifying some confusing statements. I appreciate this is an article particularly focused on the survey results and is part of a set with the 2020 a and b articles referred to. Perhaps some intro statements to help readers refer more clearly back to those and the context would help, especially if they come across this one first, as it's easy to see it as a bit orphaned.

Thank you for your constructive review. We appreciate your time and effort to make the manuscript better.

**Specific feedback comments:**

The Introduction section is quite lengthy and your study does not really get going until line 74. While the context of holding this event is really important, plenty of other publications explore the specifics of the effects of Nepalese earthquakes on communities and infrastructure. The discussion on infrastructure in particular here seems out of place when students and their families cannot control this directly by their awareness. I would recommend drawing out the key points directly relevant to your study, refer to key articles and reduce significantly.

We intend to drive the readers starting from the current risk status in terms of economics and status, and we describe the importance of education in reducing the seismic risk in the introduction. We also describe the current status of Nepal for earthquake preparedness, risk level, and cite some work that has been done earlier in the country by different groups. As the main objective of the exhibition is to make people aware of earthquakes and prepare them for future earthquakes, we mentioned some background information that could motivate readers to delve into the depth of the manuscript.

Repetition in line 85-86, could be combined above into line 80-81.

The repeated text is merged in lines 80-81.

Figure 1 is not necessary and not introduced in text. If you really want to use it, it could be combined with Figure 2, which I note is also not introduced in text at this point. Figure 2 would illustrate the point well by itself. Should be referred to in line 87.

We have moved Figure 1 to the Supplementary Material, and we have cited Figure 2 in the text line 87.

Lines 87-89 - need to mention the seismograph exhibition here in this list so can refer to Figure 3.

The jumping test exhibition is added, and Figure 3 is referred to in the suggested lines.

Lines 115-119 - largely repeats paragraph 85-90. I would recommend deleting or combining.

We have combined the content of lines 115-119 with lines 85-90.

Really nice presentation of the modules.

Thank you very much for your appreciation.

Lines 267-268 - It's not clear here whether the surveys done in 2018 or 2020 were done before or after an activity or exhibition without reading the 2020b article. For this direct comparison in this study, where you show both before and after, it would be worth clarifying those parameters here as it matters for frame of reference.

We have added information about the surveys done in 2018 and 2020, and clearly mentioned that the surveys were done for similar objectives and before the exhibition.

Lines 274-275 - it's a little unclear here whether 2000 people responded to the survey and only 495 picked at random to analyse, or whether only 495 people were picked to be surveyed. Saying "responded" implies that all were offered the chance to survey but only 495 did so. Which are students out of the 2000? Are all students? Just needs a couple of clarifying statements.

We hosted approximately 2,000 visitors during the one-day exhibition, of whom 495 students participated in the pre-exhibition survey. Due to space and time limitations, it was not feasible to survey all attendees; however, participants were selected randomly, ensuring representation from all participating schools. Additional details about the surveyed individuals—such as occupation and age—are provided in Supplementary Figures S10 and S11, where more than 95% of respondents are identified as students. The manuscript text has been updated to reflect this information.

Line 286 - needs a statement here like "These changes are outlined in the following sections for specific questions, also surveyed in 2018 and 2020 for direct comparison".

Thank you for the suggestion. We added the statement in the mentioned line.

Lines 303-304 - I think this statement needs to be clarified. There is not such notable improvement before and after the exhibition, but there is a longer term trend that is notable where you compare to the other years. At the moment this statement up front implies just looking at the exhibition effect.

We have removed the 'notable' word from the sentence and modified the sentence in terms of trend pattern.

Figures 5 to 9 need to use a different colouring and/or symbology. Having different colours for all when you are comparing two different timelines at once (comparing years against a before and after same year), muddles or obscures your results. I would recommend having the 2025 data before and after the same colour and different tone or symbology, alternatively, a gap between each year with the two 2025 results paired, could also work.

As the figures are intended to demonstrate the impact of the exhibition on changes in participants' knowledge, awareness, and preparedness regarding earthquakes, we acknowledge that color or symbol modifications could be made. However, to maintain visual consistency and ensure clear comparison across surveys, we have chosen to use distinct colors for each survey instead. Introducing gaps between the years would make the figures overly cluttered and reduce the clarity of information presentation.

Line 316 - Please delete the first statement. It's a likelihood, not a fact.

OK. The first statement is deleted.

Line 317 - "exceeding" should be "equal to or exceeding". Refer to Figure 6 at the end of this statement.

Thank you. The sentence corrected and figure referred.

Lines 331-334 - It's unfortunate that the exhibition itself was included in this question as it makes your data less clear. It's a very odd question to ask. I would leave this question out. Compare to other years, as that's the interesting data, not the exhibition in the same year.

We acknowledge that some changes in the numbers are due to participants who had never attended any disaster risk education training selecting "no" in the pre-survey and "yes" in the post-exhibition survey. Compared with earlier results, the number of people engaged in earthquake-related training has increased, not only through our activities but also through programs organized by other institutions such as the Red Cross, local governments, NGOs, and INGOs. We prefer to keep data from all years to illustrate the temporal evolution on this topic.

Figure 7 - minor spelling mistake on graph. Missing an s " I know earthquakes cannot be prevented".

Sorry for the typo. The figure is updated.

Lines 366-367 - I would just delete the last statement as it doesn't really add anything.

The sentence is deleted.

Line 374 - How has 68% changed to 80%? See line 316. Also refer back to Figure 6. Do you need to repeat this here?

The correct number is 68% and is corrected in the text. We prefer to keep one sentence here as this tells about the probability of hazards within the topic of risk.

Figure 9 - I would recommend switching the Low and High to the opposite side of the graph - consistent with other questions format.

OK. The figure is modified as suggested.

Figure 10 - Again, I would rearrange the order of the bars here. Q19 results on the right should be ordered from No to Yes, significantly, left to right, not right to left.

The figure is modified by rearranging the order of the bars.

Page 21 - nice section!

Thank you.

Line 467 - delete "the fact"

**Words deleted.**

Really nice set of recommendations and I agree a follow up survey and or programmes to evaluate whether students followed through with their intentions around communication would be excellent. Like the other referees have suggested, combining these into the previous would make it stronger.

We have combined three sections into one and modified the text as commented by other referees.